# Comment on “Black Hole Entropy: A Closer Look”

**DOI:** 10.3390/e22101110

**Published:** 2020-10-01

**Authors:** Pedro Pessoa, Bruno Arderucio Costa

**Affiliations:** 1Department of Physics, University at Albany (SUNY), Albany, NY 12222, USA; 2Department of Physics and Astronomy, University of British Columbia, Vancouver, BC V6T 1Z4, Canada

**Keywords:** black holes, entropy, non-additive entropies, thermodynamics, inference

## Abstract

In a recent paper (Entropy 2020, 22(1), 17), Tsallis states that entropy—as in Shannon or Kullback–Leiber’s definitions—is inadequate to interpret black hole entropy and suggests that a new non-additive functional should take the role of entropy. Here we counterargue by explaining the important distinction between the properties of extensivity and additivity; the latter is fundamental for entropy, while the former is a property of particular thermodynamical systems that is not expected for black holes. We also point out other debatable statements in his analysis of black hole entropy.

## 1. Introduction

In his paper, Tsallis [1] presents some questionable statements on black hole entropy. He affirms that since Bekeinstein–Hawking (BH) entropy is not proportional to the black hole volume, it would imply that Boltzmann–Gibbs (BG) entropy would be inappropriate to describe black holes. He then proposes a different non-additive functional meant to replace entropy.

This article is organized to rebuke this idea. In Section 2, we present the foundations for entropy and show how the additive entropy functional does not necessarily lead to extensivity. In Section 3, we revisit the laws of black hole thermodynamics and how they can be accurately accounted for on the basis of additive entropy.

## 2. Entropic Foundations

The work of Jaynes [2,3] solidified the ideas of Boltzmann and Gibbs on the foundations of statistical physics by designing the method of maximum entropy. Probability distributions p(x) should be selected from the maximization of entropy
(1)$$S\left[p|q\right]=-\int dx\;p\left(x\right)\ln\frac{p(x)}{q(x)}\;,$$
under constraints meant to represent the relevant information for the problem at hand. Here, q(x) is a prior distribution generally taken to be uniform. Similarly, in the context of density matrices in quantum mechanics, the entropy for a density matrix $\hat{\rho }$ based on a prior density $\hat{\varphi }$ is given by the Umegaki entropy
(2)$$S\left[\hat{\rho }|\hat{\varphi }\right]=-\mathrm{Tr}\left[\hat{\rho }\ln\hat{\rho }-\hat{\rho }\ln\hat{\varphi }\right]\;,$$
where Tr represents the trace of the matrix.

The form of entropy is deeply attached to fundamental concepts of statistical inference. Since the work of Shannon [4], there has been deep foundational research [5,6,7,8] on the criteria that an entropy functional should obey. A modern understanding [8] is that entropy should abide by two design criteria (DC) that can be roughly explained as: DC1 *subdomain independence*—local information should have only local effects; and DC2 *subsystem independence*—that a priori independent subsystems should remain independent, unless the constraints explicitly require otherwise. The unique functional that fits these criteria is (Equation 1), also called Kullback–Leiber (KL) entropy. Tsallis calls (Equation 1), and several particular cases of it, BG entropies.

A consequence of DC2 that can be directly seen from (Equation 1) is that entropy is *additive*. That means if a state *x* can be written as a composition of two substates, $x=\left(x_{1},x_{2}\right)\in X=X_{1}\times X_{2}$, and the subsystems are statistically independent, $q\left(x\right)=q\left(x_{1}\right)q\left(x_{2}\right)$ and $p\left(x\right)=p\left(x_{1}\right)p\left(x_{2}\right)$, the entropy for the composite system is the sum of the entropies for each subsystem,
(3)$$S\left[p\right]=-\int dx_{1}\;dx_{2}\;p\left(x_{1}\right)p\left(x_{2}\right)\ln\frac{p\left(x_{1}\right)p\left(x_{2}\right)}{q\left(x_{1}\right)q\left(x_{2}\right)}=-\int dx_{1}\;p\left(x_{1}\right)\ln\frac{p(x_{1})}{q(x_{1})}-\int dx_{2}\;p\left(x_{2}\right)\ln\frac{p(x_{2})}{q(x_{2})}\;.$$

The probabilities in statistical physics are obtained from the maximization of (Equation 1) after the imposition of constraints in the form of expected values
(4)$$\int dx\;p\left(x\right)a^{i}\left(x\right)=A^{i}\;,$$
for a set of real functions $a^{i}\left(x\right)\in \left\{a^{1}\left(x\right),a^{2}\left(x\right),\ldots ,a^{k}\left(x\right)\right\}$—referred to as sufficient statistics. To avoid confusion, we will use upper indexes to refer to elements in the set of sufficient statistics as in (Equation 4) and lower indexes for elements of the set of subsystems, as in (Equation 3). For example, internal energy, total volume, and magnetization are identified as constants $A^{i}$ in (Equation 4).

Since maximization of the entropy under the set of values $A=(A^{1},A^{2},\ldots ,A^{k})$ and the prior q(x) determines a unique posterior, we are led into writing the chosen distribution as p(x|A) and the maximum value for entropy as a function (rather than a functional) of A, S(A)=S[p(x|A)]. As explained by Jaynes [9], in a system *x* that evolves in time under a Hamiltonian dynamics S(A) is guaranteed not to reduce; therefore, S(A) is identified as the thermodynamical entropy. This result relies only on the fact that S(A) is constructed from the entropy functional (Equation 1). A similar function constructed from the maximization of another functional will *not* be compatible with the laws of thermodynamics.

In thermodynamics, one typically is interested in properties of the system only in the thermodynamic limit. If the system in equilibrium is homogeneous, one can divide it into N≫1 identical parts, which are small compared to the full system but large enough so that the thermodynamic limit is still descriptive of the individual parts. In well-studied applications of statistical physics (e.g., Van der Waals gas [10] and the Ising model [11]), the interactions between constituents are short-ranged, meaning that if the *N* subsystems are large enough, each sufficient statistics can be approximated as a sum of functions calculated separately for the subsystems.

In the Appendix A, we show how this condition leads to independent probabilities for each subsystem and also how the quantities $A^{i}$ and the thermodynamical entropy S(A), seen as functions of *N*, are homogeneous of degree 1 and are therefore extensive. In this description, extensivity, as a property of the thermodynamical entropy of some systems, is not fundamental nor a direct consequence of the additivity of entropy. Rather, it is a property that emerges from approximations in the sufficient statistics as the method of maximum entropy is applied.

Tsallis claims:

“ We frequently verify perplexity by various authors that the entropy of a black hole appears to be proportional to its [black hole’s] area whereas it was expected to be proportional to its volume. Such perplexity is tacitly or explicitly based on a sort of belief (i) that a black hole is a three-dimensional system, and (ii) that its thermodynamic entropy is to be understood as its Boltzmann–Gibbs (BG) one. ”

It first should be noted that it is unclear how Tsallis defines the internal volume *V* of a black hole. We nevertheless agree with the assertion that BH entropy is not proportional to *V*. For a Schwarzschild black hole, the BH entropy is proportional to the square of its mass *M*, and we do not see any meaningful definition of $V\propto M^{2}$. But this comes as no surprise for two independent reasons. First, it is much more usual in statistical physics to constrain the expected value of the total energy than the volume. For a black hole, *M* and *V* are not proportional to each other, meaning that *even if* the entropy were proportional to the total energy, it could still fail to be proportional to the “volume” of the black hole.

Second, a system bound by self-interaction is a clear example of violation of the criteria for statistical independence explained before. Such systems require a long-range attractive interaction between their parts, so the energy of the total system is necessarily considerably smaller than the sum of the energies of its parts. This suggests that the entropy of a black hole cannot be expected to be proportional to its mass or volume.

Thus, when Tsallis claims:

“It is our standpoint that, whenever the additive entropic functional $S_{BG}$ is thermodynamically non-extensive, the entropic functional to be used for all thermodynamical issues is not given by Equations (1)–(4), but by a non-additive one instead”,

His argument is flawed because as the entropy for a system happens not to be extensive, it does not mean the entropy functional needs to be changed to a non-additive one. As explained, assigning probabilities using a functional that differs from (Equation 1) violates the DC, which is unjustifiable. If subsystem independence is not guaranteed by the inference procedure, subsystems are going to be correlated even in the total absence of information that indicate so. Fundamentally, it means that no description of a system will be adequate without knowledge of every other physical system, even those known to not interact with the system of interest. In addition, as commented, the arguments that identify S(A) as the thermodynamical entropy are independent of extensivity [9], which incidentally is not expected for a black hole.

To conclude our discussion of the foundations of entropy, we want to refer the reader to authors who have reported that (i) substituting entropy by a non-additive functional leads to inconsistent statistics [12,13,14,15], as expected from the violation of the DC and (ii) that distributions once thought to arise only from these non-additive functionals can, indeed, be found by maximizing (Equation 1) [16,17,18,19,20], confirming, from different perspectives, that (Equation 1) is the appropriate form for entropy.

## 3. Black Hole Specifics

In his article, Tsallis presents questionable statements about the entropy of black holes. Plenty of references are provided to the (correct) claim that, in semiclassical, standard general relativity, the entropy of a black hole is given by the BH formula, which in natural units [21] is written as
(5)$$S_{\mathrm{BH}}=\frac{A}{4},$$
where *A* is the area of a section of the event horizon. However, Tsallis claims that

“In all such papers it is tacitly assumed that $S_{BG}$ [BH formula] is the thermodynamic entropy of the black hole. But, as argued in [24,26], there is no reason at all for this being a correct assumption.”

Unlike this claim, black holes were found to satisfy the laws of thermodynamics if assigned with an entropy given by (Equation 5) without appealing to arguments coming from probabilities or statistical mechanics. At present, we are not aware of any direct, explicit calculation starting from (Equation 1) leading to expression (Equation 5) in the literature without additional assumptions. In the context of theories of quantum gravity, progress in this direction has been made, e.g., Strominger and Vafa in string theory [22] or Lewkowycz and Maldacena in the holographic context [23], but it must be emphasized that these arguments are *not* the reason why (Equation 5) is widely accepted as the black hole’s entropy.

Rather, black holes are assigned with an entropy thanks to the combination of two key findings. First, the laws of black hole mechanics are mathematically analogous to the laws of thermodynamics [24]. Namely, the constancy of its surface gravity κ across the horizon of a stationary black hole is analogous to the zeroth law of thermodynamics. The analogue of the first law is the constraint variations δ the parameters of stationary black holes obey, Ref. [24] $\alpha \kappa \delta \left(\frac{A}{8\pi \alpha }\right)=\delta M$, where *M* is the mass of a black hole, which for simplicity is assumed to be static and charge-free, and α is an arbitrary constant. The second law is analogous to the area theorem, a general result that implies that the area of sections of the event horizon cannot decrease as one moves to the future under a general class of assumptions. It is important to point out that the laws of black hole mechanics were derived from classical general relativity alone, without any reference whatsoever to arguments coming from statistical mechanics.

The second ingredient is Hawking’s discovery that the analogue of temperature in these laws, ακ, is precisely the physical temperature of the Hawking radiation when $\alpha =\frac{1}{2\pi }$ [25]. This radiation is a consequence of effects of quantum mechanical origin. Since *M* also possesses a physical meaning as the energy of the black hole, the laws of black hole mechanics were elevated to having thermodynamic significance [26]. Furthermore, the identification of α as above pinned the black hole entropy to the expression (Equation 5). But at this point, no derivation came from first principles of statistical mechanics and no calculation of entropy from (Equation 1) leading to (Equation 5) was necessary for the conclusion to hold.

The independent claim:

“In what concerns thermodynamics, the spatial dimensionality of a (3+1) black hole depends on whether its bulk (inside its event horizon or boundary) has or not non negligible amount of matter oranalogous *(sic)* physical information.”

Cannot be sustained. Neither the classical laws of black hole mechanics [24,27] nor the derivation of the Hawking effect [25,28,29] assumes a particular form (2- or 3-dimensional) for the distribution of classical matter inside the black hole. More than that, it is justifiable to assign a non-zero entropy to an eternal black hole as in the maximally extended Schwarzschild solution [29,30], which has zero classical matter everywhere except at the singularities (which are not part of the classical spacetime, where the calculations assigning an entropy to the black hole are carried out). This strongly suggests that any attempt to frame black hole entropy in common grounds with the entropy of ordinary matter should not rely on assigning an entropy density to the matter.

In fact, as recently shown by one of us [31], general properties of the entropy functional as in (Equation 2) originate both the first and second law of black hole thermodynamics in semiclassical gravity. Hence, per Section 2, there are common grounds on which one can understand the origins of both ordinary and black hole thermodynamics. On these grounds, entropy is associated with the information one has about the system. Therefore, we do not see the need to move away from this definition and interpretation of entropy.

For black holes, the existence of an event horizon prevents an external observer from knowing the initial data for quantum fields. Roughly speaking, variations of the entropy (Equation 2) associated with the reduced state of free quantum fields obtained by taking the partial trace over these inaccessible degrees of freedom generate all of the variations of entropy appearing in the laws of black hole thermodynamics [31].

## 4. Conclusions

We contested many arguments from Tsallis’ paper. From statistical physics considerations, the fact that BH entropy is not proportional to the black hole “volume” is not indicative of an issue. The substitution of KL entropy by a non-additive functional violates principles of statistical inference and is not necessary to explain the existence of non-extensive thermodynamical entropies. Finally, there are solid foundations in support of the interpretation of the laws of black hole thermodynamics from the usual additive entropy.

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
