# Peer review of "Comment on Tsallis, C. Black Hole Entropy: A Closer Look. Entropy 2020, 22, 17"

_entropy, 2020, doi:10.3390/e22101110_

Round 1

Reviewer 1 Report

Dear Authors,

I am attaching my set of comments.

Author Response

We can see that the anonymous reviewer gave comprehensive comments to our paper. We are thankful for the time taken to check and fully understand each of our arguments and how they fit as replies to Tsallis' article and gave interesting suggestions on how to improve. Since they decided not to sign the review, we added a acknowledgment to an anonymous reviewer, but, if they request, we can change it to their name.

We are also thankful for the reviewer's comments on our writing, although we are both based on English-speaking countries neither of us is a native speaker. Because of this, their suggestions and corrections as specific comments were extremely valuable.

A summary of how the reviewer's suggestions were adopted is attached.

Reviewer 2 Report

Nice paper.  No need for changes.

Author Response

We thank the reviewer for the support.

Reviewer 3 Report

The authors (Pessoa and Costa) are critiquing a prior paper ("Black Hole Entropy: a Closer Look", author Tsallis).  The current authors correctly identify that the motivations of Tsallis for considering the non-Boltzmann Gibbs measure proposed in "a Closer Look" are suspect, and that the entropy measure proposed there does not satisfy expected requirements for a thermodynamic entropy (as the present authors correctly explain).

This Comment is already clearly written and correct in its arguments, so I only have a few comments  to add.

First, I also am not aware of a derivation of black hole entropy beginning from a probability distribution paradigm as in eqn (1); there is an accounting of black hole entropy from a microscopic counting of degrees of freedom (its most famous example is Strominger, A.; Vafa, C. Microscopic origin of the Bekenstein-Hawking entropy. Phys. Lett. B 1996, 379, 99;  which is reference [12] in the Tsallis paper; that accounting is in the context of a class of supersymmetric black holes but there is a very large amount of further work regarding progressively less supersymmetric systems, although not yet the non-supersymmetric black holes we see in our sky).

Separately, there is also an accounting for black hole entropy due to holography and the Ryu-Takayangi principle, which itself is an extremization principle; I am not aware of any interpretation of this extremization principle directly as over a class of probability distributions, however it is possible it has been discussed; the most closely related work I am aware of is

1) Lewkowycz, A; Maldacena, J. Generalized Gravitational Entropy, JHEP 08 (2013) 090

2) Harlow, D; The Ryu-Takayangi Formula from Quantum Error Correction (Comm. Math Phys 354, 865-912(2017)

In addition to the possible relevance of the line of extremization work mentioned above, there is also another criticism of the Tsallis work which the present authors may wish to include.  The entropic measure Tsallis proposes has some features in common with Renyi entropy, which while not equivalent to the thermodynamic entropy can indeed provide more information about a quantum system than the thermodynamic entropy alone.  Accordingly, both Renyi entropy and another motivated variation on it have already been extensively studied in the context of black hole information.  So there already exists a well-studied, well-supported alternative measure, which does provide important information about gravitational systems beyond the thermodynamic entropy; there is thus no need to propose another measure which is significantly less well-motivated. The most well-known reference is

3) Dong, X.  The Gravity Dual of Renyi Entropy, Nature Communications 7, Article number 12472 (2016).

Again, the present authors' criticism of the Tsallis paper is already well-founded and well-supported and can be published as-is, but if the authors wish to include further criticism along the lines above that would be appropriate.

Author Response

We are thankful for the reviewers' comments to our work. Before addressing the particular points raised we would like to say that an acknowledgment to the anonymous reviewer was added to the final revised version.

A brief discussion of extant calculations of black hole entropy coming from quantum gravity was added in section 3 starting at line 113. It was also emphasized that there is no need to appeal to anything above the semiclassical level to account for black hole thermodynamics.

Reviewer 4 Report

Attached please find the review report.

Author Response

We gladly received the anonymous reviewer's comprehensive comments to our paper. We are thankful for the time taken to check and fully understand each of our arguments and how they fit as replies to Tsallis' article. Since they decided not to sign the review, we added a acknowledgment to an anonymous reviewers.

We would like to say that we are also thankful for the reviewer's comments on our writing, although we are both based on English-speaking countries neither of us is a native speaker. Because of this, their suggestions and corrections as specific comments were extremely valuable.

Here we address the particular points raised by the reviewer:

[1] The paragraph starting on line 71 was entirely rephrased.

[2] As the reviewer suggested not necessary, we do not give a direct answer to this point. However in its current form we believe that the paper makes clear that the BH entropy is also not expected to be proportional to mass (see new sentence starting at line 82).

[3] Sentence in line 126 was completely reformulated.

[4] That was a poor construction on our part. "For similar reasons" was replaced by "The independent claim" see line 136.

[5-6]Nomenclature ``Kurskal-Szekeres'' was replaced by ``maximally extended'' Schwarzschild solution. Added the qualifier ``except at the singularity'' on line 144.

[7] We corrected the mistake and we verified the data for each particular reference.

Also, we found in our friends and family some volunteers to help with the English editing, one of them, as suggested, was "without prior knowledge on the subject matter".
Several other minor text corrections were performed. The reviewer can see them written in red in this new iteration of the manuscript.

Round 2

Reviewer 1 Report

Dear Authors,

I only have a minor comment: Please, double check the word "oranalogous" in line 132.

Author Response

We are thankful for the reviewer work.

We checked that the 'oranalogous' was in the  commented article. We believe the author meant 'or analogous' as we can't see the word 'oranalogous' being defined in any English dictionary. Although this is understandable and over all irrelevant, we decided to keep in in our paper on the ground that whenever we quote Tsallis' work this quotation should be made ipsis litteris, we don't want to force an interpretation on Tsallis' words.